# Complexes of 2-Amino-3-methylpyridine and 2-Amino-4-methylbenzothiazole with Ag(I) and Cu(II): Structure and Biological Applications

**Muhammad Hanif** [1], **Awal Noor** [2], **Mian Muhammad** [1], **Farhat Ullah** [3], **Muhammad Nawaz Tahir** [4], **Gul Shahzada Khan** [5] and **Ezzat Khan** [1,5,*]

1    Department of Chemistry, University of Malakand, Chakdara 18800, Khyber Pakhtunkhwa, Pakistan; hanifkhan939@gmail.com (M.H.)
2    Department of Basic Sciences, Preparatory Year Deanship, King Faisal University, Al-Hassa 31982, Saudi Arabia; anoor@kfu.edu.sa
3    Department of Pharmacy, University of Malakand, Chakdara 18800, Khyber Pakhtunkhwa, Pakistan
4    Department of Physics, University of Sargodha, Sargodha 40100, Punjab, Pakistan
5    Department of Chemistry, College of Science, University of Bahrain, Main Campus, Sakhir 32038, Bahrain
*    Correspondence: ekhan@uom.edu.pk

**Abstract:** Coordination complexes (**1**–**4**) of 2-amino-4-methylbenzothiazole and 2-amino-3-methylpyridine with $Cu(CH_3COO)_2$ and $AgNO_3$ were prepared and characterized by UV/Vis and FT-IR spectroscopy. The molecular structure for single crystals of silver complexes (**2** and **4**) were determined by X-ray diffraction. The coordination complex (**2**) is monoclinic with space group *P*21/c, wherein two ligands are coordinated to a metal ion, affording distorted trigonal geometry around the central Ag metal ion. The efficient nucleophilic center, i.e., the endocyclic nitrogen of the organic ligand, binds to the silver metal. Ligands are coordinated to adopt *cis* arrangement, predominantly due to steric reasons. The O(2) and O(3) atoms of the $NO_3^-$ group further play an important role in such type of ligand arrangement by hydrogen bonding with the $NH_2$ group of ligands. Complex (**4**) is orthorhombic, *P*212121, comprising two molecules of 2-amino-3-methylpyridine as ligand coordinated with the metal ion, affording a polymeric structure. The coordination behavior of the ligand is identical to that in complex **2**, wherein ring nitrogen is coordinated to the metal center and bridged to another metal ion through an $NH_2$ group. The resulting product is polymeric in nature with the Ag metal in the backbone and ligand as the bridge. Compounds (**2**–**4**) were found to be luminescent, while **1** did not show such activity. All compounds were screened for their preliminary biological activities such as antibacterial, antioxidant and enzyme inhibition. Compounds exhibited moderate activity in these tests.

**Keywords:** 2-amino-3-methylpyridine; 2-amino-4-methylbenzothiazole; copper complexes; silver complexes; coordination polymers; biological applications

## 1. Introduction

Coordination chemistry plays an important role in daily life and has tremendously contributed over the last several decades [1–3]. The applicability of a complex depends on the type and oxidation state of the metal, and the nature of the attached ligand. The nature of the ligand affects the properties of metal complexes; a slight change in the structure of an organic molecule as ligand can alter the properties of its metal complex to a large extent, particularly biological efficiency. Recently, several drugs (metal-based) have been explored with excellent in vitro results; however, in vivo results show certain limitations, and this is one of the reasons that very few metal-based drugs are recommended for clinical trials. The challenges for chemists are to explore the in vivo stability and the compatibility of metallodrugs within biological systems, which depends on the selection of organic

molecules as ligand and metal ions [4,5]. The applications of coordination complexes also depend on the denticity of the attached ligand and on the coordination environment around the central metal [6,7]. Literature studies show that complexes bearing coordination number 4 (square planner) are more effective against various diseases, particularly cancer. The *cis*-isomer has been found to be an effective chemotherapeutic agent but not the *trans*-isomer, such as cisplatin. The *cis/trans* arrangement greatly affects the efficiency, application and reactivity of coordination complexes [8,9]. The biological proficiency of free ligand can differ from its respective coordination complexes. Mostly, the biological activities of coordination complexes have been found to be higher than those of uncoordinated free ligands [10]. Currently, coordination complexes of copper(I/II) and silver(I) are a focus of research, owing to their wide range of biological applications [11]. A large number of these complexes have been reported, with promising antibacterial [12], antifungal, anticancer [13], antitumor [14,15] and antiproliferative activities [16,17].

Many heterocycles, particularly pyridines and thiazole-containing compounds, occupy an important place in coordination chemistry because of their flexible bioactivities due to their multifunctionality [18]. The 2-aminothiazole derivatives are an important class of organic compounds bearing very good biological activities and applications in drug development. Benzothiazole derivatives have photosensitizing properties, and have been an integral component of dyes for a long time [19,20]. Thiazole derivatives bear potential ligating sites and are of particular importance due to their multi-coordination modes and their significance in the pharmaceutical and medicinal fields. They exhibit antimicrobial, antidegenerative, anti-inflammatory, anti-tumour, anti-HIV, antihypertensive and cytotoxic activities, being reflected by a large number of drugs on the market containing this moiety [21–23].

Keeping in view the importance of coordination complexes, herein we report the preparation, characterization, luminescence behavior and biological studies (antibacterial, antioxidant and enzyme inhibition) of four coordination complexes. Complexes were obtained by the reaction between 2-amino-4-methylbenzothiazole and 2-amino-3-methylpyridine with metal salts of $Cu(CH_3COO)_2 \cdot H_2O$ and $AgNO_3$.

## 2. Results and Discussion

The structures of compounds **1–4** are shown in Scheme 1. Copper complexes are amorphous powders, while silver complexes are crystalline in nature. Single crystals of good quality for complexes **2** and **4** were isolated and X-ray diffraction data were collected. All complexes were studied with the help of UV-visible and FT-IR spectroscopy.

**Scheme 1.** Proposed structures of compounds **1–4**.

### 2.1. UV-Visible Spectroscopic Studies of Complexes **1–4**

The spectra of compounds **1–4** are given in Figures S1 and S2, and the respective absorbance values are summarized in Table 1. Two bands in the spectrum of each complex 230, 292 (**1**), 230, 264 (**2**), 238, 292 (**3**) and 244, 296 nm (**4**) correspond to $\pi \rightarrow \pi^*$ intra-ligand electronic transitions. According to the literature, the $\pi \rightarrow \pi^*$ transition appearing in the

range of 200–300 nm shifts to a longer wavelength upon complexation, while the n→π* transitions of the free ligand disappear in complexes [24]. The absence of n→π* transition bands at the region of 300–400 nm in all compounds indicate coordination through the non-bonding (n) electron pair of N.

**Table 1.** Absorption in nm ($\lambda_{max}$, π→π*) and the respective molar absorptivity ($\varepsilon_{max}$, dm³.mol⁻¹.cm⁻¹) values of compounds **1–4**.

| Compound | $\lambda_{max}$ | $\varepsilon_{max}$ |
|---|---|---|
| **1** | 230 | – |
| | 292 | – |
| **2** | 230 | 1,754,123 |
| | 264 | 2,382,020 |
| **3** | 238 | 923,295 |
| | 292 | 1402,698 |
| **4** | 240 | 344,745 |
| | 296 | 650,567 |

*2.2. FT-IR Spectra of Compounds **1–4***

Literature studies reveal that NH stretching appears as a typically broad signal in FT-IR spectra in the range of 3500–3200 cm⁻¹ [25,26]. The peak at 3409, 3281 in compound **1**, 3414, 3282 in compound **2**, 3337, 3204 in compound **3** and 3398, 3342 cm⁻¹ in compound **4** correspond to stretching vibration of the NH₂ group. C-H aromatic stretching appears at 3066 (**1**), 3123 (**2**), 3089 (**3**) and at 3153 cm⁻¹ in compound **4**. Similarly, CH stretching of CH₃ appears at 2944, 2943, 2911 and 3123 cm⁻¹ in compounds **1–4**, respectively [27]. The peaks at 2316, 1626 in **1**, 2325, 1627 in **2**, 2459, 1609 in **3** and 2325, 1626 cm⁻¹ in compound **4** are assigned to CN (C-N and C=N) stretching. The band corresponding to C=C stretching appears at 1520, 1521, 1551 and 1520 cm⁻¹ in compounds **1–4**, respectively [28,29]. Peaks at 649 and at 647 cm⁻¹ in **1**, **2** are predominantly because of C-S-C stretching, respectively [30,31]. A relatively weak FT-IR band at 470, 471, 478 and 470 cm⁻¹ in compound **1–4**, respectively can be assigned to Cu-N and Ag-N stretching vibrations. Similarly, the weak bands 563, 554 in **1**, **3** can be assigned to Cu-O stretching [32–34]. The appearance of these weak FT-IR bands for Cu-O, Ag-N stretching confirm the formation of the proposed complexes (spectra are provided in Supporting File, Figure S3–S6).

*2.3. XRD Structural Description of Complexes **2** and **4***

Structure refinements and solution parameters for complexes **2** and **4** are summarized in Table 2 and important structural features are given in Table 3. The molecular structure of complex **2** is given in Figure 1. The crystal is monoclinic with space group $P2_{1/c}$, wherein two ligands are attached to the central metal ion (Ag) in addition to nitrate ($ONO_2^-$), affording a distorted trigonal complex. The efficient nucleophilic site (endocyclic nitrogen) of the ligand coordinates to the Ag(I) ion. The two organic ligands are arranged in a trans manner with respect to each other across the metal ion, while the counter anion $ONO_2^-$ is also directly attached to the metal ion through O(1). The three coordinated angles around metal ion are N(1)-Ag(1)-O(1), N(2)-Ag(1)-O(1) and N(1)-Ag(1)-N(3) are 99.20, 99.35 and 158.65°, respectively. The expected L-Ag-L bond angle deviates from planarity due to the additional coordination of the $ONO_2^-$ ion, as has been reported for a structurally analogous complex [35]. The plane of both organic ligands is oriented at right angles with respect to each other across the metal ion, which causes the interaction to minimize. The distance between organic ligand and metal center is slightly different i.e., 2.207 and 2.192 Å, Ag(1)-N(1) and Ag(1)-N(3), respectively. Intramolecular H-bonding between O(1) and amine H₂N (interatomic distance O(1) . . . N(2) 3.021 Å) in the molecule plays a pivotal role in hindering free rotation of one ligand. The O(2) and O(3) of the nitrate ion are exclusively involved in intermolecular H-bonding. The O(3) is simultaneously linked to N(4), S(1) and N(2) of neighboring molecules with separation distances of 3.059, 3.119 and 2.952 Å, respectively.

Through these secondary interactions, three molecules associate together. S(1) and N(2) belong to the same molecule, while N(4) is from another molecule. Among the most prominent interactions of O(2) is the O(2) . . . S(2), with a separation distance of 3.157 Å. These H-bonding interactions stabilize the 2D supramolecular structure along axes *c* and *a*. Further short-ranged interactions include CH . . . S, $\pi$-$\pi$, etc., to strengthen the association between molecules in solid state. The N-O bond distances of nitrate ions are different i.e., N(5)-O(2) 1.219, N(5)-O(1) 1.248 and N(5)-O(3) 1.242 Å. While the longest distance is a clear indication of coordination of O(1) with an Ag ion, the shortest bond length reveals that the electron density responsible for double bonding is highly concentrated between N and O(2).

**Table 2.** Summarized parameters pertaining to crystal structure solution and refinement of compounds **2** and **4**.

| Compound No. | 2 | 4 |
|---|---|---|
| Formula weight | 498.33 | 527.02 |
| Empirical formula | $C_{16}H_{16}N_4O_3S_2Ag$ | $C_6H_8N_4O_4Ag_3$ |
| Temperature (K) | 296 | |
| Wave length (Å) | 0.71073 | |
| Space group | $P2_{1/c}$ | $P2_12_12_1$ |
| Crystal system | Monoclinic | Orthorhombic |
| a(Å) | 11.147 (2) | 5.2969 (3) |
| b(Å) | 12.488 (3) | 8.8341 (4) |
| c(Å) | 13.491 (2) | 18.6386 (12) |
| Z | 4 | |
| Volume ($A^3$) | 1801.3 (6) | 872.16 (8) |
| Density (Mgm$^{-3}$) | 1.838 | 2.117 |
| wR($F^2$) | 0.077 | 0.087 |
| $R\,[F^2 > 2\delta(F^2)]$ | 0.031 | 0.033 |
| Theta (max) | 27.0 | 29.9 |
| (h, k, l)max | (14, 14, 17) | (6, 10, 16) |
| (h, k, l)min | (−14, −15, −16) | (−5, −11, −24) |
| R (reflection) | 3887 | 2038 |
| F (000) | 1000 | 544 |
| μ (mm$^{-1}$) | 1.38 | 2.29 |
| Goodness of Fit | 1.03 | 1.04 |

**Table 3.** Summarized structural parameters, bond angles (°) and bond lengths (Å) for compounds **2** and **4**.

| Compound 2 Bond Lengths | | Compound 4 Bond Lengths | |
|---|---|---|---|
| N(1)-Ag(1) | 2.207 (2) | Ag(1)-N(1) | 2.218 (5) |
| N(3)-Ag(1) | 2.192 (2) | Ag(1)-O(1) | 2.494 (4) |
| S(1)-C(1) | 1.751 (3) | Ag(1)-O(2) | 2.348 (5) |
| S(1)-C(3) | 1.735 (2) | O(1)-N(3) | 1.242 (6) |
| N(1)-C(1) | 1.311 (3) | O(2)-N(3) | 1.256 (6) |
| N(1)-C(2) | 1.397 (3) | O(3)-N(3) | 1.233 (7) |
| N(3)-C(9) | 1.304 (3) | N(1)-C(1) | 1.339 (7) |
| Bond Angles | | Bond Angles | |
| N(1)-Ag(1)-N(3) | 158.65 (7) | N(1)-Ag(1)-O(2) | 143.81 (16) |
| C(1)-N(1)-Ag(1) | 121.99 (16) | N(1)-Ag(1)-O(1) | 127.61 (15) |
| C(2)-N(1)-Ag(1) | 122.26 (15) | O(2)-Ag(1)-O(1) | 77.39 (14) |
| C(9)-N(3)-Ag(1) | 121.57 (17) | N(3)-O(1)-Ag(1) | 109.8 (3) |
| C(10)-N(3)-Ag(1) | 127.41 (16) | O(3)-N(3)-O(2) | 119.4 (5) |
| C(3)-S(1)-C(1) | 89.05 (12) | O(3)-N(3)-O(1) | 121.0 (5) |
| C(11)-S(2)-C(9) | 89.27 (12) | O(1)-N(3)-O(2) | 119.6 (5) |

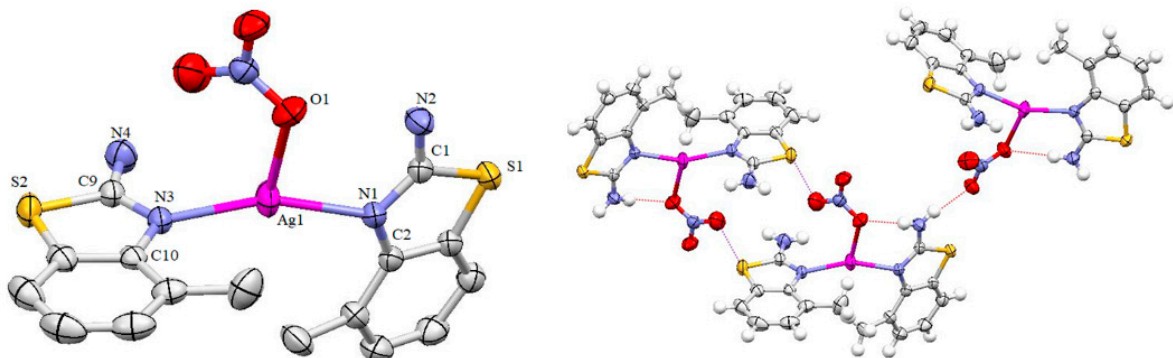

**Figure 1.** Solid state structure of **2** along with partial atomic numbering schemes (**left**), intermolecular H-bonding in molecules of the complex (**right**). Thermal ellipsoids are drawn at 50% probability level; H atoms are omitted in the molecule; color scheme: oxygen (red), nitrogen (light blue), sulfur (yellow), silver metal ion (purple) and carbon (grey).

The Hirshfeld surface and fingerprint plots were calculated to visualize the intermolecular contacts present within the crystal structure of compound **2**. The corresponding acceptor and donor atoms showing C—H···O and N—H···O intermolecular hydrogen bonds are shown as bright red spots on the Hirshfeld surface (Figure 2). The H···H interactions (31.6%) appear as the largest contributing region on the Hirshfeld surfaces (Figure 3). The C···H/H···C contacts (20.4%) also contribute predominantly to the Hirshfeld area. The pair of sharp spikes with a contribution of 17.9% represents the O···H/H···O contacts due to the intermolecular C—H···O and N—H···O hydrogen bonding (2.028 and 2.269 Å). The interactions due to S···H/H···S contacts are depicted as symmetrical wings with a contribution of 11.8%. Other interatomic contacts that have significant contribution to crystal packing as compared to other interatomic contacts are N···H/H···N and C···C π interactions, with percentage contributions of 5.7% and 3.1%, respectively. The interatomic contacts that have comparatively less contribution to crystal packing are S···O (1.8%), O···C (1.5%), Ag···S (1.3%), N···C (1.0%), S···S (0.9%), C···S (0.9%), S···N (0.8%), N···O (0.5%) and Ag···O (0.3%) interactions.

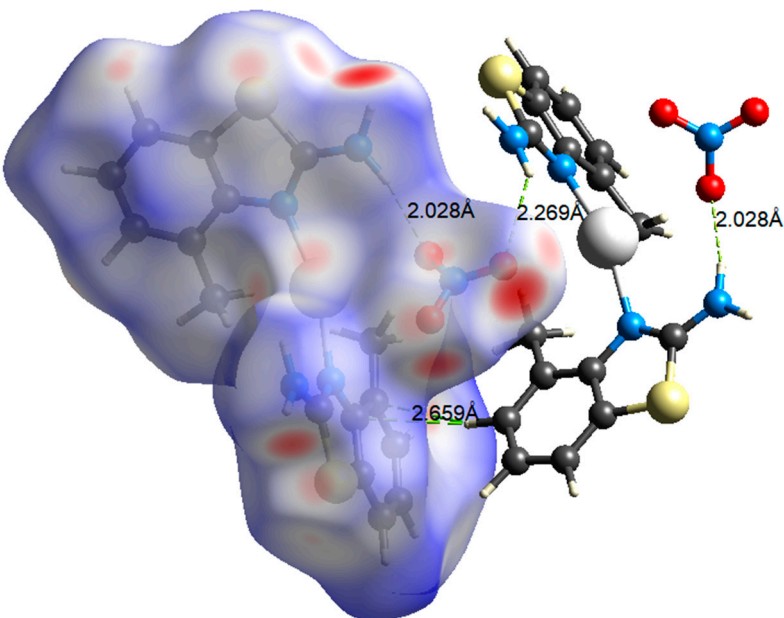

**Figure 2.** An overview of 3D Hirshfeld surface of complex **2**, plotted over $d_{norm}$ in the range of −0.4462 to 1.3360. Light green dotted lines signify prominent hydrogen bonds.

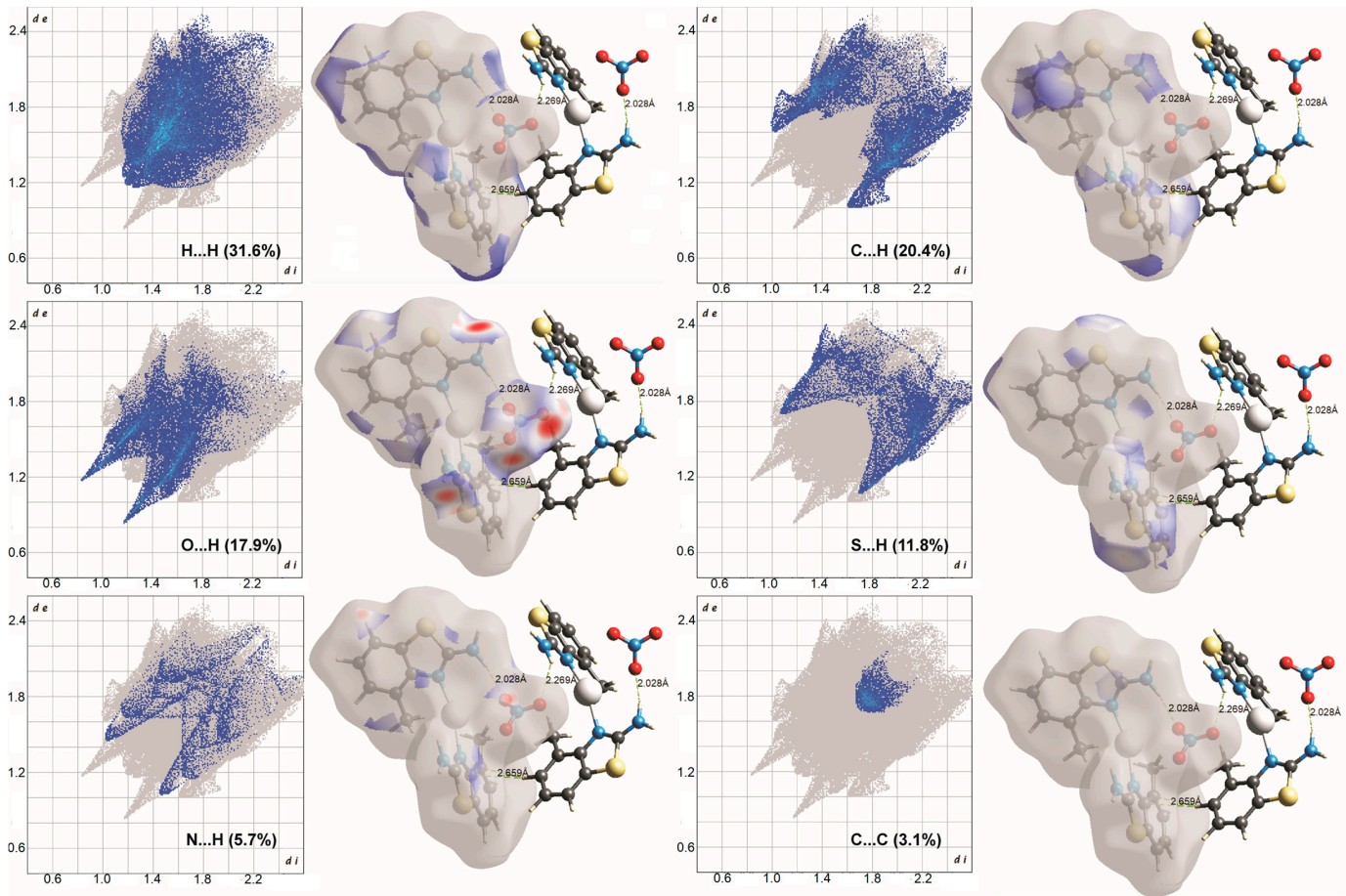

**Figure 3.** A 2D finger plots showing intermolecular interactions in complex **2**, percent contribution for each contact is specified.

### 2.4. Structure Description of Compound **4**

Complex **4** is orthorhombic with space group $P2_12_12_1$, where two molecules of the ligand (2-amino-3-methylpyridine) are coordinated to the metal ion, affording a polymeric structure (Figure 4). The structure of the molecule indicates that a reaction occurs between metal and ligand in equimolar amounts. Ligand is linked to the metal ion with the help of a strong nucleophilic N center (pyridine), as has been observed in complex **2** and other related complexes [36–39]. In the subject complex, the ligand acted in a slightly different way than the expected. The ligand bridges two metal ions together through its endocyclic (N) and exocyclic (NH$_2$) N-atoms, thus affording the formation of a polymeric structure. The Ag(1)-N(1) is 2.218 Å, which is slightly longer than the Ag-N bond in complex **2**. In the extended structure of complex **4**, the main role has been played by the nitrate anion. The geometry around the metal ion is distorted tetrahedral, with bond angles N(1)-Ag(1)-O(2), N(1)-Ag(1)-O(1), N(1)-Ag(1)-N(2), N(2)-Ag(1)-O(1) and O(1)-Ag(1)-O(2) around silver metal atom of 143.81, 127.59, 109.17 82.87 and 77.40°, respectively. The difference in bond distances between the nitrogen and oxygen atoms of nitrate groups [N(3)-O(1), N(3)-O(2) and N(3)-O(3) 1.242, 1.256 and 1.233 Å] could be attributed to unequal participation of the O atoms in non-covalent interactions. In complex **4**, various non-covalent interactions are observed, namely O$_2$NO···H-NH, Ag···O, Ag···N, Ag···H, Ag···C, H$_2$CH···O and O$_2$NO···H-C, which stabilize the 3D structure of the complex.

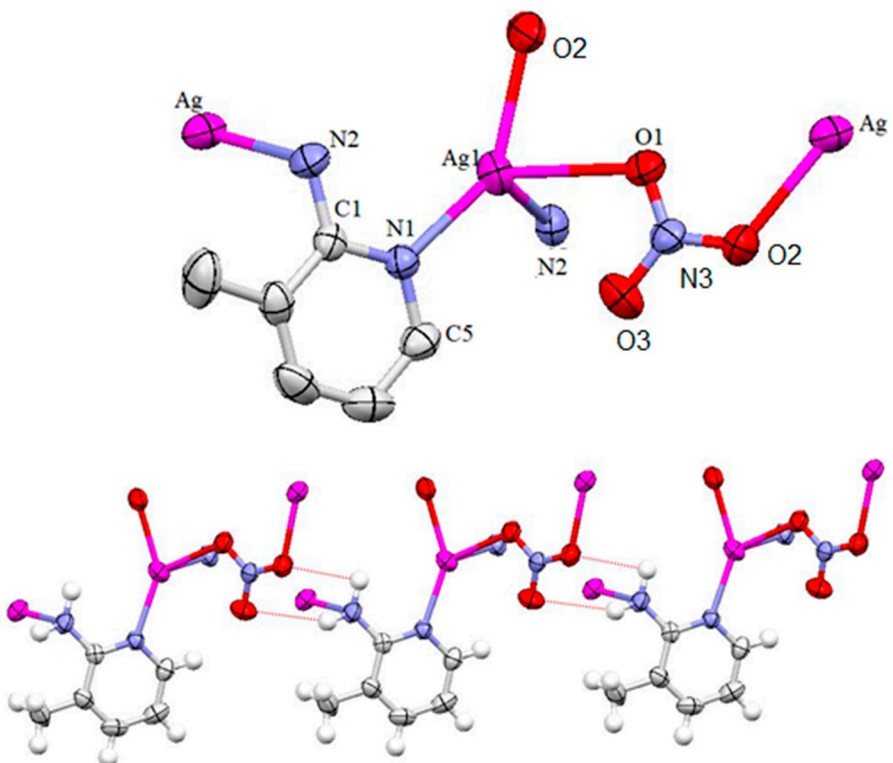

**Figure 4.** Solid state structure of **4** with partial atomic numbering schemes; thermal ellipsoids are drawn at 50% probability; hydrogen atoms are omitted for clarity reasons (upper structure). Intermolecular hydrogen bonding is shown in lower structure, causing extension of the structure in a 1D fashion; color scheme is the same as for complex **2**.

The nitrate anion plays an important role in stabilizing the 2D structure of the polymeric structure along *a* and *b* axes, as shown in Figure 5. The O(1) and O(2) of the nitrate anion make a bridge between two adjacent silver(I) ions. The organic ligand 2-amino-3-methylpyrine also acts as a bridging ligand between two metal ions. The distance between O(1) . . . Ag(I) and O(2) . . . Ag(I) is 2.494 and 2.348 Å, respectively. Similarly, the distance between N(1) . . . Ag(1) and N(2) . . . Ag(1) is 2.218 and 2.668 Å, respectively. The difference in N-Ag bond lengths is obvious, owing to the soft and hard nature of both the ligating sites and the soft metal ion. The soft center, i.e. the ring nitrogen atom, binds firmly as compared to the $NH_2$. The Ag-N bond lengths are longer in complex **4** as compared to the literature data [40,41]. Ag . . . Ag interactions in compounds discussed above were not observed due to the coordinatively saturated nature of the Ag ion, in comparison to some reported complexes [42–44]. It is obvious from the literature reports dealing with Ag complexes that the coordination sphere of the metal ion and nature of the ligand attached to it is key to the supramolecular structure and variety of secondary interactions.

There are two red spots on the Hirshfeld surface (Figure 6), indicating H-bonding contacts due to C—H···O and N—H···O interactions (2.373 and 2.443 Å, respectively). Like compound **2**, H···H interactions (30.0%) contribute the most to the Hirshfeld surfaces; however, unlike **2**, the contribution due to the O···H/H···O (24.0%) and the C···H/H···C (15.7%) contacts are swapped (Figure 7). The interatomic contacts due to N···H/H···N interactions (5.6%) in **4** are comparable to those in **2**, followed by a significant contribution from Ag···H/H···Ag contacts (5.0%) appearing as two symmetrical peaks in the two-dimensional finger plots. Other interactions that involve Ag atoms are Ag···O/O···Ag (3.9%), Ag···N/N···Ag (3.2%) and Ag···C/C···Ag (2.0%) contacts. The O···C/C···O (4.7%) and O···N/N···O (2.3%) interactions in **4** are also more pronounced than those in **2**. The interatomic contacts due to N···C (1.0%) and C···C (0.5%) π interactions contribute the least to the crystal packing.

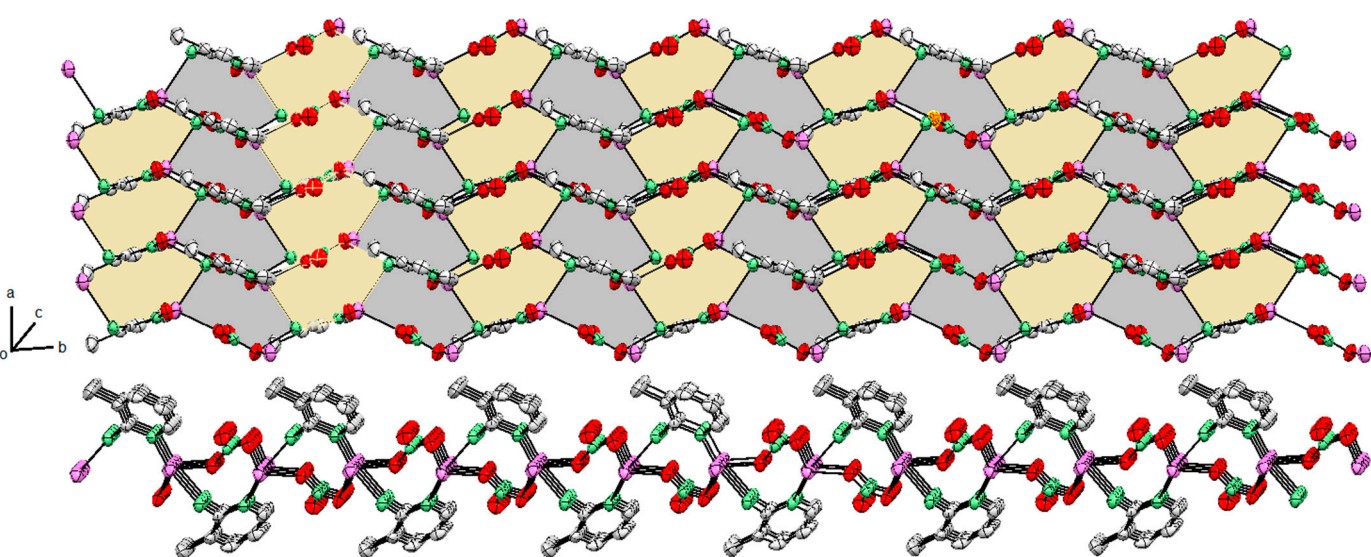

**Figure 5.** 2D polymeric structure (different views) of the complex 4, extended along a, b through nitrate and 2-aminopyridine moieties.

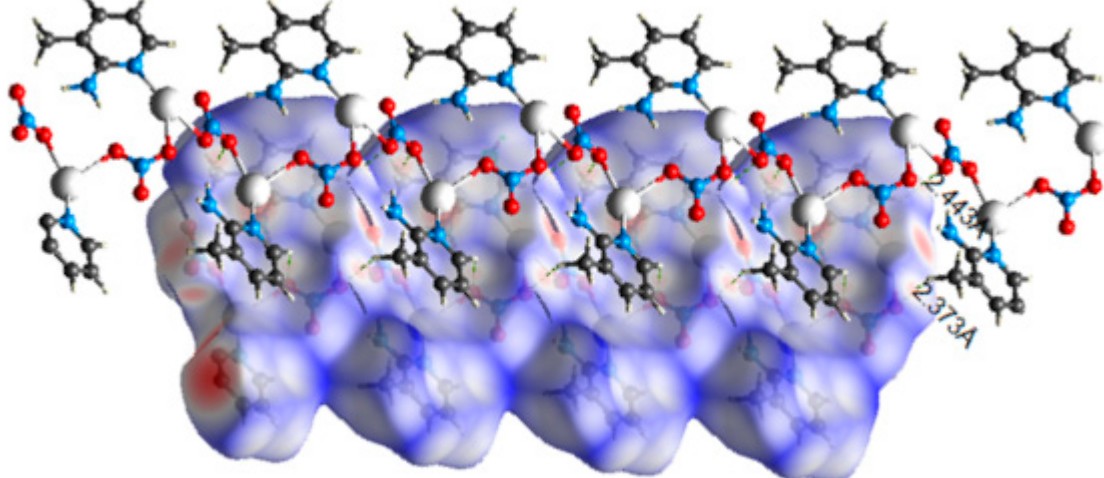

**Figure 6.** View of the three-dimensional Hirshfeld surface of **4**, plotted over $d_{norm}$ in the range of −1.1545 to 1.4419.

*2.5. Antibacterial Activity*

In developing countries, one-third of total deaths are caused by infectious diseases. Infections caused by multidrug resistance (MDR) are of serious concern and need attention to be addressed in an efficient way. There is a need to design and develop efficient antibacterial drugs against MDR, particularly with advanced mechanisms of action [45,46]. The coordination complexes **1–4** were tested against representative gram-positive and gram-negative bacterial strains (*K. pneumonia*, *Streptococcus*). The activities of complexes are compared with standard drugs, and zones of inhibition are represented in Table S1. The data reveal that among four complexes, **2** is more effective and shows ZI 16 and 18 mm against *K. pneumonia* and *Streptococcus*, respectively. The ZI values obtained for complex **2** are very close to standard cefixime and azithromycin. Complex **4** is next to exhibit considerable activity against *Streptococcus* and moderate activity against *K. pneumonia*. The ZIs of compound **4** against *Streptococcus* and *K. pneumonia* were found to be 16 and 9 mm, respectively. Compound **3** also showed promising activity against *Streptococcus* with a ZI of 16 mm and least activity against *K. pneumonia* (ZI 7 mm).

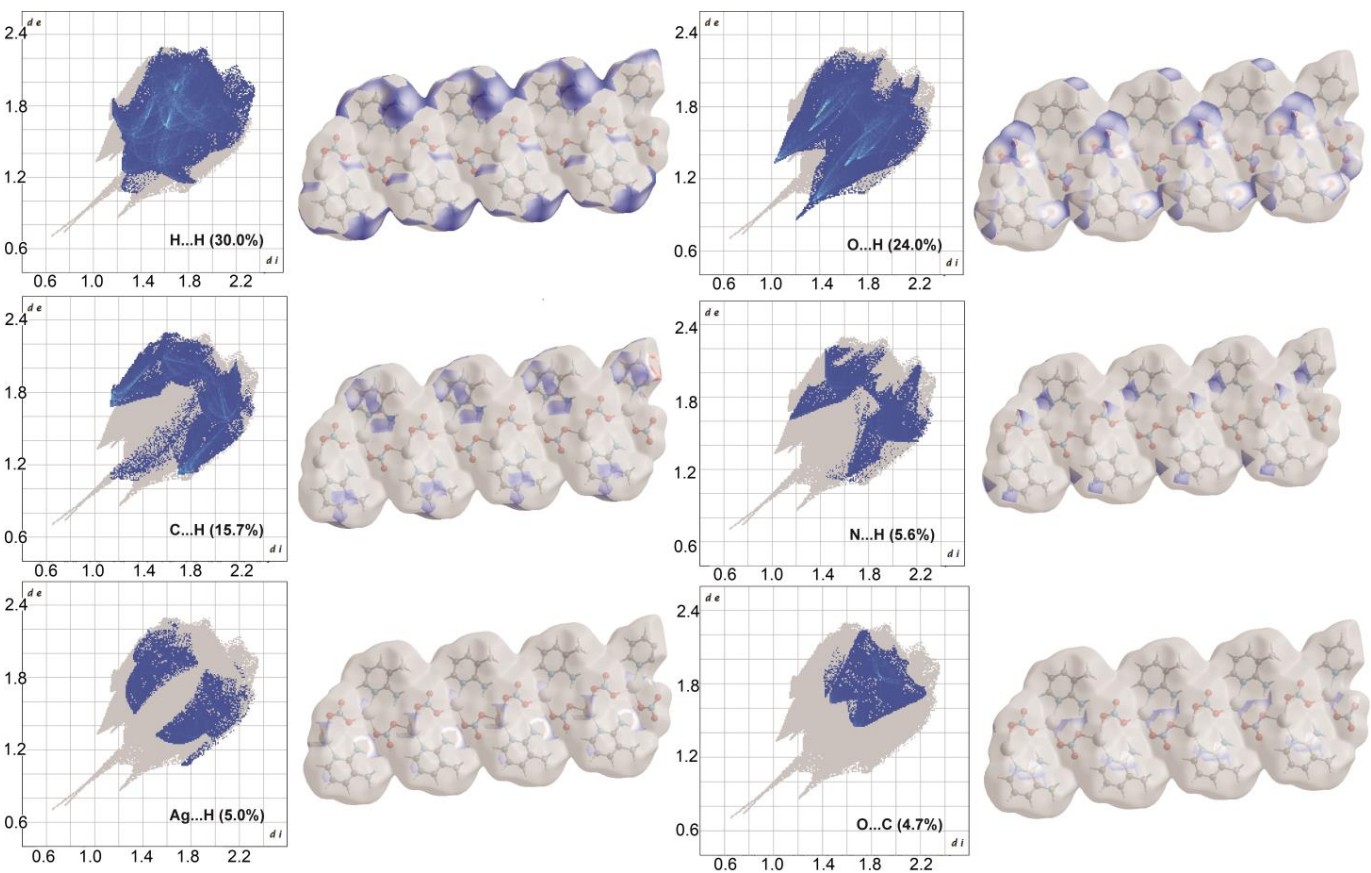

**Figure 7.** 2D finger plots for **4** showing intermolecular contacts. The percentage of contribution is also specified.

### 2.6. Free Radical Scavenger Studies

Continuous formation of free radicals causes chronic diseases, and their timely control within the body is necessary [47,48]. Various coordination complexes have been reported as efficient antioxidant agents, and scientists are still in search of more efficient and biocompatible antioxidants, particularly towards commercialization [36,49]. During this study, the antioxidant efficiency of complexes was investigated against DPPH and ABTS. Antioxidant efficiency was governed by a decrease in the resultant absorbance of reference. The antioxidant activities of coordination complexes **1–4** are summarized in Table S2, where the copper-containing complex **1** exhibits comparatively high activity against ABTS and DPPH (IC$_{50}$ values of 199.10 and 150.88 µg/mL, respectively). However, the efficiency of the compound is far lower than the activity of standard ascorbic acid and further research on different derivatives must be continued.

### 2.7. Enzyme Inhibition Studies of Compounds

Alzheimer's disease (AD) is a dementia caused by irregular production of acetylcholinesterase (AChE) and butyrylcholinesterase (BChE) in the body [36,39,50,51]. The AChE and BChE inhibitory activities of compounds are summarized in Table S3. Complex **1** came out with the highest activity against AChE and BChE (IC$_{50}$ 0.95, 0.87 µg/mL, respectively). The IC$_{50}$ values of the complex are close to that of the standard galantamine and can be a potential candidate for future studies.

### 2.8. Luminescence Study of Complexes **1–4**

Keeping in view the importance of luminescence, all the complexes discussed in this article were tested for their luminescence activity. Compounds **2**, **3** and **4** are luminescent, their λ$_{max}$ values of excitation and emission are given in Table 4, and the respective spectra

are given in Figures S7–S9. Compounds **2**, **3** and **4** exhibit strong luminescence emissions at 337, 369 and 352 nm, and excitations at 311, 335 and 295 nm, respectively. Peak intensity and good resolution of excitation and emission peaks invite further investigation. These compounds have additional functional groups which can be used for further applications, particularly sensing applications [51,52].

**Table 4.** The $\lambda_{max}$ values of excitation and emission for compounds **2**–**4**.

| Compound | $\lambda_{max}$ Excitation (nm) | $\lambda_{max}$ Emission (nm) |
| --- | --- | --- |
| 2 | 311 | 337 |
| 3 | 335 | 369 |
| 4 | 295 | 352 |

## 3. Materials and Methods

All chemicals used in this work are commercial products (TCI, Tokyo Japan) and were used without further purification (2-amino-4-methylbenzothiazole, 2-amino-3-methylpyridine, copper(II) acetate and silver nitrate). All reactions were carried out under aerobic conditions and the melting points (uncorrected) of the compounds were determined in sealed capillary tubes by Stuart SMP-10, Japan. The FT-IR spectral data were collected by SHIMADZU FT-IR model 18,400 within the range 4000–400 cm$^{-1}$. The single crystal X-ray diffraction data of compounds **2** and **4** were collected by using a Bruker kappa APEXII CCD diffractometer at ambient temperature. Crystal structure solution and refinements were accomplished by using SIR97, WinGX31, PLATON and SHELXL97 [53–57].

### 3.1. Hirshfeld Surface Analysis of Crystalline Complexes

Hirshfeld surface analysis of the crystalline compounds was carried out using the same procedure as reported earlier [58]. The Hirshfeld surfaces and 2D fingerprint plots of both the complexes were generated using Crystal Explorer 17.50 [59]. For input files, X-ray single-crystal crystallographic information files (CIFs) were used. Interactions within the molecules were determined. Interactions shorter than the sum of the corresponding van der Waals radii of the atoms, and the longer contacts with the positive $d_{norm}$ value are highlighted by red and blue spots, respectively.

### 3.2. Luminescence, Antibacterial, Antioxidant and Enzyme Inhibition Studies of **1**–**4**

Methanolic solutions with varying concentrations of complexes **1**–**4** were incubated for 1 h at ambient temperature. The compounds were screened for their luminescence activity (fluorescence spectrophotometer, model RF-5301) in the range 200–800 nm.

Selected bacterial strains were considered in this study, namely, *Streptococcus* and *K. pneumoniae*, and screening for this purpose was carried out by well assay and using the MIC method [60]. Muller-Hinton Ager culture media was set accordingly, while media, borer, petri dishes and pipettes were sterilized at 121 °C with 15 psi pressure for 15 min. Microbial cultures of 10$^6$ CFU were applied on each petri dish, and 6 mm bores were made with a sterile borer. A volume of 200 μL of each compound was applied and the resultant system was incubated for 24 h at 37 °C. The zone of inhibition (ZI) for each compound was manually noted, and the available standards cefixime (CEF) and azithromycin were used as reference drugs against *K. pneumonia* and *Streptococcus*, respectively. The temperature of agar (Muller Hinton) was maintained at 45 °C, and the standard and compounds were applied to separate petri dishes within the range of 0.07–1000 μg/mL. On the media, the strains were incubated at 37 °C for 24 h and the respective results were obviously noted. The organic compounds (ligands) were also tested for their efficacy against the selected strains.

DPPH (2,2-Diphenyl-1-picrylhydrazyl), a stable and robust free radical, was used in determining the scavenging efficiency of compounds **1**–**4** [61]. Changes in the colour of methanolic solutions of DPPH after addition of compounds was taken as an indication of antioxidant potential, followed by quantification with UV-visible spectroscopic measure-

ments. An amount of 20 mg of DPPH in 100 mL MeOH was prepared as stock solution, a 3 mL portion of this solution was taken, and its absorbance was adjusted (0.75 at 517 nm), hereafter termed control solution. The stock solution was covered with aluminium foil to avoid interaction with light during the process of free radical formation and was retained for 24 h. A solution of each compound was prepared by dissolving 5 mg compound in 5 mL methanol. The concentration range of working solutions for this study was maintained between 62.5 and 1000 mg/mL (1000, 500, 250, 125, 62.5). From each working solution, 2 mL was mixed with the same volume of DPPH solution and the contents were incubated for 15 min in the dark. The efficiency for each sample was calculated according to the literature procedure [58].

The enzyme inhibition (AChE and BChE) potentials of compounds **1**–**4** were explored using the Ellman assay [62]. Solutions were prepared according to the literature procedure; they were incubated for the optimum time, and absorbance of samples was determined at 412 nm for around 4 min at $30 \pm 1\,^{\circ}C$. The inhibition of compounds **1**–**4** and the standard was calculated in percent from their absorption with respect to change in time [63].

*3.3. Synthesis of Coordination Complexes (**1**–**4**)*

Complexes **1**–**4** were synthesized by following the literature procedure [35,64]. Ligand solution was slowly added over a period of 5 min into the corresponding metal salt solution in appropriate molar ratios and the reaction mixtures were stirred overnight. Crystals of complexes were grown in the same solvent by slow evaporation of the respective solution at ambient temperature. For the preparation of complex **1**, ligand 2-amino-4-methylbenzothiazole (0.088 g, 0.440 mmol) in 10 mL methanol was added to copper acetate (0.144 g, 0.440 mmol) solution in the same solvent, keeping a metal to ligand ratio of 2:1, respectively. The color of the reaction mixture changed to green and the reaction contents were stirred overnight. For complex **2**, a solution of 2-amino-4-methylbenzothiazole (0.040 g, 0.243 mmol) was treated dropwise with a solution of $AgNO_3$ (0.02 g, 0.121 mmol) in THF. Complex **3** was obtained as a result of mixing 2-amino-3-methylpyradine (0.850 mL, 0.912 g, 8.433 mmol) with $Cu(CH_3COO)_2$ (0.841 g, 0.425 mmol) in methanol. After overnight stirring, green precipitates were formed, solid was separated and dissolved in chloroform, and crystals were obtained at room temperature after 24 h. Complex **4** was prepared by mixing equimolar amounts of ligand (2-amino-3-methylpyradine) and $AgNO_3$, following the same procedure. Crystals were obtained and their melting points and other parameters were measured.

Compound (**1**) yield = 65%; UV spectral bands $\lambda_{max}$ (nm) = 230, 292 nm; FT-IR $\nu(cm^{-1})$ = 3409, 3281 (NH), 3066 (CH), 2944 (CH), 2316 (CN), 1626 (C=N), 1520 (C=C), 649 (C-S), 563 (Cu-O), 470 (Cu-N).

Compound (**2**) yield = 68%; 166–167 $^{\circ}C$; UV spectral bands $\lambda_{max}$ (nm) = 230, 264; FT-IR $\nu(cm^{-1})$ = 3415, 3282 (NH), 3123(CH), 2943 (CH), 2325 (CN), 1627 (C=C), 1521 (C=N), 647 (C-S), 471 (Ag-N).

Compound (**3**) yield = 65%; UV spectral bands $\lambda_{max}$ (nm) = 238, 292; FT-IR $\nu(cm^{-1})$ = 3337, 3204 (NH), 3089 (CH), 2911 (CH), 2459 (CN), 1609 (C=N), 1551 (C=C), 554 (Cu-O), 478 (Ag-N).

Compound (**4**): yield = 69%; m.p = 190–192 $^{\circ}C$; UV spectral bands $\lambda_{max}$ (nm) = 240, 296; FT-IR $\nu(cm^{-1})$ = 3398(NH), 3342 (CH), 3153 (CH), 2325 (CN), 1626 (C=C), 1520 (C=N), 470 (Ag-N).

## 4. Conclusions

Pyridine-based ligands which are simple in structure, such as 2-amino-4-methylbenzothiazole and 2-amino-3-methylpyridine, are efficient in bonding with Cu(II) and Ag(I). Reactions afford desired complexes in reasonable amounts. Ag(I) complexes were isolated in crystalline form, wherein the central metal ion is distorted trigonal. In both the complexes, nitrate ions plays an important role in stabilizing the respective complex in solid state. The $NH_2$ group in 2-aminopyridine was able to bridge two silver ions together

and a polymeric structure was thus afforded. Complexes were tested for their antibacterial efficiency against representative gram-positive/negative strains. A silver complex [Ag(2-amino-4-methylbenzothiazole)$_2$]NO$_3$ was efficient against both *Streptococcus* and *K. pneumonia*, with observed ZI values of 16 and 18 mm in comparison to the standard 18 and 19.5 mm, respectively. Enzyme inhibition of all the complexes was not encouraging and further derivatives need to be explored in future work. Luminescence studies of the complexes give well-resolved $\lambda_{em}$ (337–369 nm) and $\lambda_{ex}$ (295–335 nm), complex [Cu(2-amino-4-methylbenzothiazole)$_2$(CH$_3$COO)$_2$].

**Supplementary Materials:** The following supporting information can be downloaded at: https://www.mdpi.com/article/10.3390/inorganics11040152/s1. Crystallographic data for the structural analysis have been deposited with the Cambridge Crystallographic data center, CCDC Nos. 2239388 and 2239389 for 2 and 4, respectively. May be obtained free of charge from the +44-(1223)*336-033 or Email: deposit@ccdc.cam.ac.uk or www: http://www.ccdc.cam.ac.uk. Supporting files contains UV-visible (Figure S1 UV-visible spectra of copper complexes **1** and **3**; Figure S2 UV-visible spectra of silver complexes **2** and **4**), FT-IR (Figure S3–S6 FT-IR spectra of compound **1–4**, respectively) and luminescence spectra (Figure S7–S9 Luminescence spectra of compound **2**, **3** and **4**, respectively) of complexes reported in this article. Tables contain summarized data of various tests i.e., Table S1 (Antibacterial zones of inhibition measured in mm for compounds **1–4**); Table S2 (Antioxidant potentials for compounds **1–4**) and Table S3 (IC$_{50}$ values of compounds as AChE and BChE enzyme inhibitors).

**Author Contributions:** Conceptualization, M.H. and E.K.; Funding acquisition, E.K.; Investigation, M.H. and E.K.; Methodology, M.H. and E.K.; Project administration, E.K.; Resources, E.K.; Supervision, E.K.; Validation, E.K.; Visualization, M.H., A.N., M.M., F.U., M.N.T., G.S.K. and E.K.; Writing–original draft, M.H., A.N., G.S.K. and E.K.; Writing–review & editing, M.H., A.N. and E.K. All authors have read and agreed to the published version of the manuscript.

**Funding:** This work was supported by the Deanship of Scientific Research, Vice Presidency for Graduate Studies and Scientific Research, King Faisal University, Saudi Arabia [Grant No. 3179].

**Data Availability Statement:** Data is contained within the article or Supplementary Material.

**Acknowledgments:** This work was supported by the Deanship of Scientific Research, Vice Presidency for Graduate Studies and Scientific Research, King Faisal University, Saudi Arabia [Grant No. 3179]. EK is thankful to HEC, Pakistan for financial support under NRPU project No. 1488 and 7327.

**Conflicts of Interest:** The authors declare no conflict of interest.

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
