# Peer review of "Complexes of 2-Amino-3-methylpyridine and 2-Amino-4-methylbenzothiazole with Ag(I) and Cu(II): Structure and Biological Applications"

_inorganics, doi:10.3390/inorganics11040152_

Round 1

Reviewer 1 Report

The subject of the reviewed work is current and the search for new substances with specific biological properties is in the trends of contemporary coordination and bioinorganic chemistry. 

The work is written concisely and to the point and is suitable to be published in Inorganics after some minor corrections. 

1) In the main part of publication Table S1 should be inserted.

2) In figure 4, the lack of numbering of all atoms in nitrogen group while their appear in text.

3) Bridging character of nitrate group in the polymeric compound 4 should be emphasized.

4) English should be corrected.

Author Response

The subject of the reviewed work is current and the search for new substances with specific biological properties is in the trends of contemporary coordination and bioinorganic chemistry. 

The work is written concisely and to the point and is suitable to be published in Inorganics after some minor corrections. 

Response: We thank the reviewer for being very positive and encouraging evaluation of the article. The manuscript was revised in light of the suggested changed as per below details: 

  • In the main part of publication Table S1 should be inserted.

Response: Table S1 shifted to main body of the manuscript, and deleted from supporting file. 

  • In figure 4, the lack of numbering of all atoms in nitrogen group while their appear in text.

Response: All N and O atoms are numbered in the Figure.

  • Bridging character of nitrate group in the polymeric compound 4 should be emphasized.

Response: A new Figure has been included and the role of nitrate as bridging moiety has been elaborated.

4) English should be corrected.

Response: Checked and various minor corrections made.

Thank you for constructive remarks towards improvement of the article. 

Reviewer 2 Report

In this work, the authors describe a series of Cu(II) and Ag(I) complexes and coordination polymers obtained using 2-amino-4-methylbenzothiazole and 2-amino-3-methylpyridine. The new compounds were unambiguously characterized by X-ray diffractometry as well as FT-IR and absorption spectroscopy. The crystal structures were analysed by Hirshfeld’s method. Moreover, emission and biological properties were investigated. In my opinion, the reviewed work is sound, and well-written and its results noticeably contribute to organoselenium chemistry. There is no doubt that this work will be of interest to the readership of this journal. Thus, I recommend this manuscript after addressing the following concerns:

1. In the experimental part, the elemental analysis data should be added.

2. Correct the phrase “Compound (1) yield = 65 %; m.p = UV spectral bands”.

3. I recommend expanding the citation list with the following works on emissive Ag(I) polymers supported by thiazole-based and similar ligands, e.g. DOI: 10.1107/S2056989019000124, 10.1016/j.mencom.2020.11.013, 10.1016/j.molstruc.2021.131816, 10.1002/ejic.202000109, 10.2174/1874846500801010024, 10.1016/j.ica.2019.01.036, 10.1016/j.molstruc.2004.09.022.

4. Because the authors didn’t measure the emission lifetimes, it is incorrect to claim fluorescence. Thus, I recommend replacing the word “fluorescence” with “emission” or “luminescence”.

5. The data for 1 are missing in Table 1. 

Author Response

In this work, the authors describe a series of Cu(II) and Ag(I) complexes and coordination polymers obtained using 2-amino-4-methylbenzothiazole and 2-amino-3-methylpyridine. The new compounds were unambiguously characterized by X-ray diffractometry as well as FT-IR and absorption spectroscopy. The crystal structures were analysed by Hirshfeld’s method. Moreover, emission and biological properties were investigated. In my opinion, the reviewed work is sound, and well-written and its results noticeably contribute to organoselenium chemistry. There is no doubt that this work will be of interest to the readership of this journal. Thus, I recommend this manuscript after addressing the following concerns:

Response: We thank the reviewer for positive feedback, all the suggested changes have been carried out as per below details.  

  1. In the experimental part, the elemental analysis data should be added.

Response: We do not have this facility and the said data was therefore not collected. We humbly request the reviewer to relax this condition because, at the movement it is almost impossible for us to resynthsise the compounds and measure and include their data.

  1. Correct the phrase “Compound (1) yield = 65 %; m.p = UV spectral bands”.

Response: Corrected in revised version.

  1. I recommend expanding the citation list with the following works on emissive Ag(I) polymers supported by thiazole-based and similar ligands, e.g. DOI: 10.1107/S2056989019000124, 10.1016/j.mencom.2020.11.013, 10.1016/j.molstruc.2021.131816, 10.1002/ejic.202000109, 10.2174/1874846500801010024, 10.1016/j.ica.2019.01.036, 10.1016/j.molstruc.2004.09.022.

Response: The above referred reports have been cited at appropriate places in the main text.

  1. Because the authors didn’t measure the emission lifetimes, it is incorrect to claim fluorescence. Thus, I recommend replacing the word “fluorescence” with “emission” or “luminescence”.

 Response: Replaced the word “fluorescence” by “Luminescence” in the main as well as supporting file.

  1. The data for 1 are missing in Table 1.

Response: Some of the misleading data has not been presented due to some unknow reasons.  

Thank you very much for the feedback which improved the manuscript to a larger extent.